# Construction Safety Risk Assessment and Early Warning of Nearshore Tunnel Based on BIM Technology

Ping Wu [1,*], Linxi Yang [1,2,*], Wangxin Li [1,3], Jiamin Huang [1,2] and Yidong Xu [1]

1 School of Civil Engineering and Architecture, NingboTech University, Ningbo 315100, China; liwangxin@nbt.edu.cn (W.L.); xyd@nit.zju.edu.cn (Y.X.)
2 School of Civil Engineering, Chongqing Jiaotong University, Chongqing 400074, China
3 School of Civil Engineering, Lanzhou University of Technology, Lanzhou 730050, China
* Correspondence: wuping@nit.zju.edu.cn (P.W.); 17815907570@163.com (L.Y.)

**Abstract:** The challenging nature of nearshore tunnel construction environments introduces a multitude of potential hazards, consequently escalating the likelihood of incidents such as water influx. Existing construction safety risk management methodologies often depend on subjective experiences, leading to inconsistent reliability in assessment outcomes. The multifaceted nature of construction safety risk factors, their sources, and structures complicate the validation of these assessments, thus compromising their precision. Moreover, risk assessments generally occur pre-construction, leaving on-site personnel incapable of recommending pragmatic mitigation strategies based on real-time safety issues. To address these concerns, this paper introduces a construction safety risk assessment approach for nearshore tunnels based on multi-data fusion. In addressing the issue of temporal effectiveness when the conflict factor K in traditional Dempster–Shafer (DS) evidence theory nears infinity, the confidence Hellinger distance is incorporated for improvement. This is designed to accurately demonstrate the degree of conflict between two evidence chains. Subsequently, an integrated evaluation of construction safety risks for a specific nearshore tunnel in Ningbo is conducted through the calculation of similarity, support degree, and weight factors. Simultaneously, the Revit secondary development technology is utilized to visualize risk monitoring point warnings. The evaluation concludes that monitoring point K7+860 exhibits a level II risk, whereas other monitoring points maintain a normal status.

**Keywords:** nearshore tunnel; building information modeling technology; DS theory of evidence law; construction safety risk assessment; risk warning

## 1. Introduction

In recent years, the thriving advancement of infrastructure construction has significantly bolstered China's economy, evolving into a fundamental pillar of its economic structure. Correspondingly, safety incidents in construction projects have garnered increased scrutiny and have emerged as a prominent societal issue [1]. Nearshore tunnels provide distinct advantages over alternative cross-sea transportation methods due to their expediency, speed, minimal environmental footprint, and high traffic volume. However, the enduring submersion of tunnel structures in seawater exposes them to high water pressure and creates substantial technical challenges during construction. The intricate geological conditions, coupled with potential encounters with fault zones and weathered deep grooves, can lead to water ingress and seepage, thereby introducing significant safety risks [2]. Consequently, it becomes imperative to scrutinize and analyze safety risk factors in the construction of nearshore tunnels. This helps enhance nearshore tunnel construction technology, optimizes the construction management process, minimizes accident-induced losses, and provides robust foundations for future tunnel selection, design, and construction [3].

Existing construction safety risk assessment practices suffer from three primary shortcomings. First, reliance on expert analysis and scoring introduces subjectivity, leading to reduced accuracy in evaluation outcomes [4–6]. Second, conducting risk assessments prior to construction prevents on-site operators from proposing timely, appropriate risk avoidance measures based on actual engineering issues. Lastly, the unique risk indicator systems of different construction environments hinder the applicability of insights from one project to another [1,7–11]. Implementing multi-source heterogeneous data fusion technology in underground engineering construction safety risk assessment can substantially enhance the precision of evaluation results.

Owing to the intricate environments in underground engineering and tunnel construction, there are numerous hazardous sources throughout the construction process, necessitating the establishment of additional monitoring points. Consequently, the project has to manage a significant volume of real-time monitoring data. Currently, manual monitoring, despite its inefficiencies and delayed risk warning capabilities, remains the primary method for overseeing underground engineering and tunnel construction. This also augments the likelihood of risk assessment and early warning inaccuracies due to human error [12]. Utilizing building information modeling (BIM) technology, a 3D model can be constructed and analyzed using 3D roaming, animation demonstrations, and simulation construction. This allows for the timely identification of potential hazards during construction based on safety risk factors. The BIM model can facilitate intelligent detection of monitoring points in the construction area, precise positioning of safety risk zones during the monitoring process, and implementation of construction safety risk controls. Marking of unsafe locations permits digital management of the construction site [13–16]. Integrating BIM technology into safety risk management of underground engineering construction can effectively mitigate issues such as delayed data collection, transmission, and analysis, and non-intuitive and delayed risk warnings. Collins et al. [17] examined the development of an underground construction safety risk early warning system based on BIM and Internet of Things (IoT) technologies. This study established a comprehensive early warning and control system that enables real-time dynamic monitoring by consistently recording and assessing process metrics via a BIM management platform. Ding et al. [18] integrated BIM with semantic web technology to create a construction safety risk management framework within a BIM environment and elaborated on the entire workflow of construction safety risk management, encompassing risk factor identification, risk path analysis, and risk mitigation strategies. Similarly, Lou et al. [19] investigated the construction of an urban complex project, combining BIM and augmented reality (AR) technologies. They realized real-time dynamic monitoring and control system by regularly capturing and assessing key performance indicators through the BIM management platform. Moreover, AR technology was employed across three distinct phases—pre-accident, during, and post-accident—to bolster construction quality and productivity. Lu et al. [20] developed a software plug-in bridging BIM technology with safety risk data, which can automatically calculate building safety risks and help architects and structural designers quickly select the design scheme and verify the feasibility of the method through case studies.

In addition to these technological innovations, resilience plays a significant role in safety risk assessment. It refers to a system or organization's capacity to anticipate, absorb, adapt, and recover from unforeseen disruptions or events. Within the scope of safety risk assessment, resilience serves to identify system vulnerabilities, mitigate risks, and adapt to evolving conditions. It also aids in planning for recovery and in nurturing a culture of continuous learning to refine safety protocols. According to Cimellaro et al. [21], the concept of catastrophe resilience is viewed as a synthesis of knowledge from organizational and technological disciplines, ranging from social sciences and economics to seismology and earthquake engineering. Numerous presumptions and interpretations are necessary for the study of catastrophe resilience. However, the ultimate goal is to combine data from multiple fields into a single framework, resulting in a free of erroneous assumptions or preconceived concepts of risk. The authors also provided a framework that was based on

Californian hospital structures and provided an in-depth explanation and implementation of a streamlined recovery model. Within the building system itself, as well as the losses suffered by the people the system serves, this model considers both direct and indirect losses. Aven [22] made a connection between resilience and risk, positing that risk assessments could offer valuable insights into resilience evaluations, particularly by accounting for the uncertainty of potential disruptions. Yang et al. [23] delved into the quantitative facets of resilience, proposing a triple resilience definition framework founded on perturbations, functionality, and performance. This framework also accommodates the handling of uncertainties. Resilience emphasizes a system's ability to predict, absorb, adapt, and recover from disruptive situations, providing a significant concept that encompasses reliability and risk-based thinking to guarantee the safety of these complex systems. Guo et al. [24] applied the resilience theory to the safety management of three subway construction sites through analyzing the resilience essence and assessing the system's resilience using a resilience index. They employed cloud and element extension theories to establish an analytic network process (ANP) extended the cloud comprehensive model, aiming to tackle the inherent randomness and fuzziness encountered during resilience assessments at these sites. San-gaki et al. [25] established a probabilistic framework and model for determining the probability distribution of earthquake recovery indices in order to account for various uncertainties and produce recovery curves. Additionally, they suggested a probability model that was consistent with dependability techniques and included the ground motion intensity of earthquakes, building responses, structural damage, loss of functioning of buildings, recovery, and resilience. By creating elastic response curves for a typical four-story reinforced concrete moment-resisting frame building and contrasting the findings with those using conventional techniques, the authors proved the validity of their framework and model. Forcellini et al. [26] employed a probability-based vulnerability curve approach to estimate losses and introduced a novel definition of recovery models. The study also discussed the application of the proposed framework through case studies in both fixed-base and base-isolated structural systems. The existing body of research has made substantial contributions to exploring the advantages and applications of resilience in disaster management. Some studies have focused on the concept and definition of resilience, as well as its practical implementation in risk management. Others have delved into specific domains such as natural disasters and supply chain management, examining the impact of resilience on various types of risks and corresponding mitigation strategies. These studies facilitate an enhanced understanding and application of resilience concepts in the relevant fields and the evaluation of their practical benefits in diverse settings.

In this paper, multi-source information fusion technology enables the integration of on-site data, design data, and environmental data to ascertain the dynamic safety risk level of underground engineering construction. Through Revit secondary development technology, the risk level can be real-time warned on the BIM model, thereby improving the efficiency of underground engineering construction safety risk management. The feasibility of this approach is validated using a specific nearshore tunnel in Ningbo as an example. Research findings confirm that this approach can identify and assess risks throughout the engineering process, provide early warnings, and prevent accidents during nearshore tunnel construction. This study addresses critical limitations in traditional construction safety risk assessment methods, such as the reliance on expert subjectivity and the underutilization of real-time monitoring data. By employing data fusion technology, we obtain a dynamic safety risk profile at specific monitoring points, offering a more accurate depiction of safety risk levels on the construction site. Specifically, the data fusion approach collects data from various sources and types, yielding a more holistic view of safety risks. This facilitates accurate predictive analyses by uncovering hidden correlations and patterns, thereby reducing bias and errors.

Our proposed real-time monitoring and early warning visualization methodology, based on BIM, enhances the practical utility of monitoring data within BIM platforms. This tackles issues such as the lack of intuitive early warning systems and delays in early

warnings. The approach significantly augments the efficiency of safety management throughout the construction process. It also provides a robust foundation for construction safety management that benefits all project stakeholders, diminishing the likelihood of safety incidents and elevating the project's overall safety standards.

## 2. Background and Reviews of Related Studies

### 2.1. Construction Safety Risk Assessment

In the 1970s, Professor Einstein of the Massachusetts Institute of Technology pioneered the application of safety risk research in the realm of tunnel construction and underground engineering. He extensively analyzed identification, evaluation, control, and other aspects of tunnel construction safety risk, establishing a theoretical groundwork for further research in this field. Currently, the safety risk assessment of underground engineering construction mainly employs techniques such as the fuzzy analytic hierarchy process, entropy weight method, expert scoring, sensitivity analysis, and neural network analysis [27–31]. For instance, Qian and Lin [32] synthesized a decade of major advancements in safety risk management in underground engineering construction in China, utilizing expert surveys. Through analyzing typical construction safety incidents in China over the past decade, they highlighted the emerging challenges in underground engineering safety risk management. Yoo and Kim [33] formulated a theory for risk analysis and evaluation in tunnel construction based on monitoring data collected during construction, employing statistical analysis methods. Li et al. [34] examined the characteristics and mechanisms of tunnel collapse through model experiments, offering compelling evidence for selecting risk indicators for tunnel collapse. Liu et al. [35] proposed a "collapse warning value" for soft rock tunnel construction according to relevant regulations, engineering experience, and field monitoring to guarantee the safety of the construction area. Min and Einstein [36] utilized tunnel engineering risk management to assist decision-making systems in simulating resource scheduling and planning through tunnel construction. By formulating varying tunnel construction strategies, they achieved optimization of the construction process, considering events, costs, and resources. Dammyr et al. [37] summarized the construction safety risks encountered by nearshore tunnels in Norway, suggesting the use of the drilling and blasting method to reduce the likelihood of incidents, particularly for unfavorable rock formations and high water pressure risks.

Despite the broad application and relative success of the aforementioned methods, they are not without limitations. Drawbacks include a reliance on expert subjectivity, which can result in low applicability of safety risk assessment outcomes. Furthermore, the wide array of factors influencing construction safety risks and their sources complicates the validation of evaluation results, further undermining their accuracy. Moreover, risk assessments are typically conducted pre-construction, meaning on-site construction workers are unable to take appropriate risk mitigation measures in response to actual safety issues promptly. Thus, there is a potential for using deterministic multi-source information gathered during the construction process to assess inherently uncertain construction safety risks, thereby enhancing the precision of the assessment results.

### 2.2. Integration of Multi-Source Heterogeneous Data

In today's interconnected world, underpinned by advances such as ubiquitous internet access, sensor proliferation, big data, and e-commerce, the information environment shaping the development of artificial intelligence (AI) has undergone profound transformations [38,39]. In the contemporary world, continuous innovation in information technology is facilitating the deep development of digitalization, networking, and intellectualization. Digitization involves the use of digital technology to transform information into recognizable formats for easier storage, processing, and transmission. Networking refers to the interconnection of computers, communication devices, and information resources through networks to create a global platform for information sharing. Intellectualization entails

the application of AI technologies to endow machines with capabilities that mimic human cognitive functions.

Digitization has ushered in novel opportunities for technological revolution and knowledge acquisition across industries that possess the potential to access pertinent networks. The rapid growth in computational capabilities, in accordance with Moore's Law, and the increasing incorporation of microprocessors into a myriad of products and services have accelerated the expansion of interconnected systems [40]. Novel organizational structures have reshaped the traditional Fordist model into a digitally networked society, where these networks are not only crucial for interpersonal communication but also indispensable for global innovation, collaborative research and development, inventor networks, innovation hubs (such as Silicon Valley), and worldwide value chains [41]. In this evolving landscape, the continuously advancing 5G networks are progressively assuming a critical role in propelling the growth of the IoT and various other intelligent automation applications. Advancements in intelligence, including IoT, AI, autonomous vehicles, virtual reality, blockchain, and even innovations yet to be envisioned, all necessitate the rapid and responsive connectivity, along with the minimal latency, that 5G networks deliver [42]. Digitization, networking, and intellectualization are the three main trends in the development of information technology in today's world. These three trends are mutually reinforcing, and together they drive the development of information technology to a deeper level. Digitization lays the groundwork for networking and intellectualization, networking serves as the platform that enables the digitization and intellectualization processes, and intellectualization acts as the driving force behind the other two. This plays an increasingly important role in stimulating economic and social development, modernizing the national governance system and capacity, and meeting the public's escalating demands for a better life. In the context of the "Internet Plus" era, the big data platform for Internet information supervision needs to collect data from various departments and integrate unstructured data like text, images, audio, and video. This necessitates the big data platform to handle multi-source and multi-type data [43]. Information fusion involves the comprehensive analysis and processing of data from various forms and sources to aid decision-making. The overall performance of the fusion system surpasses that of the local system. There are various methods for information fusion, which vary depending on the forms and sources of the information.

Based on Dempster's research [44], the Dempster–Shafer (DS) evidence theory is proposed, which is a further development of the evidence fusion theory from Shafer [45]. The DS evidence theory is a generalization of the probability theory. Unlike the traditional probability theory, the evidence in the DS evidence theory can associate with multiple pieces of evidence. It excels at managing uncertainty, treating uncertain and incomplete information as a range of probability uncertainty and ambiguity [46]. It can amalgamate evidence from various sources and obtain a certain level of support while considering all available evidence. The support in the DS evidence theory can be interpreted as the probability of an event occurring. This theory has been incorporated into various research fields. For instance, Zhang et al. [2] proposed a novel method that integrates fuzzy matter elements, Monte Carlo simulation technology, and DS evidence theory. By synthesizing multi-source conflict evidence, the risk level of early tunnel construction-induced building damage could be perceived. Zhou et al. [47] introduced a novel approach and system to assess and manage risks during the construction process. This approach coupled risk and quality management systems, incorporating site monitoring data, design data, and construction environment data. It utilized the DS evidence theory to merge data and calculate the overall risk index. Pan et al. [48] suggested a model built upon the DS evidence theory, interval-valued fuzzy sets, and Bayesian networks to address the issue of insufficient and inaccurate data collection in risk management. Employing multi-source information fusion technology to incorporate on-site monitoring data and appraise the safety risk level of nearshore tunnel construction, this approach yielded more precise outcomes than conventional safety risk assessment methods.

*2.3. Construction Management Based on BIM Technology*

BIM is a digital and three-dimensional visualization approach. It enables the exchange and sharing of valuable information of a construction project, from initial planning and design stages to construction, and further to operational management and demolition stages. This ensures information consistency among all parties throughout the project's entire lifecycle [49].

Since its implementation in the construction industry, BIM has quickly become a primary method due to its multi-dimensional, interactive, and shared characteristics [50]. Numerous scholars, both domestic and international, have conducted studies in the fields of BIM and construction risk. For instance, Lu et al. [20] put forth a method for quantitatively assessing safety risk during the design phase of construction projects and developed a plugin that links BIM technology with safety risk data. This plugin can automatically calculate building safety risks, aiding architects and structural designers in rapidly selecting class solutions. The method's feasibility was confirmed through examples. Collins et al. [17] analyzed the establishment of an underground engineering safety risk warning system using BIM and IoT technology. This addressed problems such as high investment, extended construction periods, complicated construction environments, and numerous unpredictable safety risk factors in underground engineering construction. The system regularly recorded and tested process indicators through the BIM management platform, implementing real-time dynamic monitoring of the security risk warning system. Ding et al. [18] integrated BIM technology with web technology to create a construction safety risk management framework within a BIM environment. A case analysis elaborated the complete process of construction safety risk management, including risk factor identification, risk path reasoning, and risk prevention strategies. Kim and Liang [51–53] concentrated on scaffolding and templates, integrating these temporary structures into automated safety inspection approaches using BIM technology. By establishing a construction safety risk monitoring platform, workers' spatial movements using scaffolding were simulated and visualized. The platform has the capability to automatically detect safety risk factors associated with scaffolding operations and devise appropriate risk mitigation measures before the operation. Kim et al. [54] used BIM technology and 3D laser scanning technology to present a systematic and practical approach for assessing the dimensions and surface quality of prefabricated concrete components. Lin et al. [55] integrated internet technology with BIM technology, proposing a construction phase quality inspection and defect management system based on BIM technology. Lou et al. [19] harnessed the combination of BIM and AR technology to manage construction quality in three stages—before, during, and after safety accidents, aiming to enhance engineering quality and production efficiency in the construction industry. The visualization feature of BIM models allows for an easier understanding of risk distribution, thus facilitating the implementation of appropriate adjustments and solutions. Applying BIM and its related technologies to safety risk management of nearshore tunnel construction holds considerable prospects and significance.

In conclusion, reviewing the existing domestic and international research shows that the primary focus is on integrating quality and risk management based on information systems, quality management using BIM, risk management using BIM, and subway construction safety risk prediction through data fusion. There is an intent to move away from subjective methods of safety risk assessment. However, limited studies have been conducted on integrating BIM technology and multi-data fusion algorithms for safety risk management of submarine tunnel construction. Therefore, this article applies the improved DS evidence theory method to fuse real-time monitoring data from construction sites and calculate the comprehensive risk level. The feasibility of this method is validated using the Ningbo nearshore tunnel as an example. Simultaneously, with the assistance of Revit secondary development technology and the C sharp programming language within the Microsoft Visual Studio 2017 Net framework, risk assessment results are integrated on Revit. This integration accomplishes a three-dimensional visual display of safety risk levels and warnings for nearshore tunnel construction. The research findings affirm that this

method effectively monitors and evaluates risks during project construction, enables early capabilities, and helps prevent accidents during nearshore tunnel construction.

## 3. Improved DS Evidence Theory Method Based on the Hellinger Distance

*3.1. Traditional DS Evidence Theory Method [45,56]*

3.1.1. Basic Concepts

(1)    Basic probability allocation

The DS evidence theory's probability allocation for each hypothesis within the recognition framework is referred to as the BPA. On the recognition framework M, the BPA is an $2^M \rightarrow [0,1]$ function $M$, named the mass function, and satisfies:

$$m(\varnothing) = 0 \text{ and } \sum_{a \subseteq M} m(a) = 1$$

(2)    Trust function

The trust function, also known as the reliability function, based on BPAm in the recognition framework, is defined as:

$$\text{Bel}(a) = \sum_{b \subseteq a} m(b) \tag{1}$$

This equation implies that for a hypothesis $a$, its trust function is the sum of all assumptions that truly belong to $a$, i.e., the mass values of $b$.

(3)    Likelihood function

The likelihood function, also called the plausibility function, on the recognition framework, based on BPAm, is defined as:

$$\text{Pl}(a) = \sum_{b \cap a \neq \varnothing} m(b) \tag{2}$$

This equation suggests that for a hypothesis $a$, its likelihood function is the sum of the mass values of all hypothesis $b$ whose intersection with $a$ is not empty.

(4)    Trust interval

In the evidence theory, for a certain hypothesis $a$ in the recognition framework, the trust function and likelihood function of the hypothesis are calculated according to the BPA function to form the trust interval $[\text{Bel}(a), \text{ Pl}(a)]$.

3.1.2. Dempster Synthesis Rules

(1)    Composition rules of two basic probability allocation functions

For $\forall a \subseteq M$, the Dempster synthesis rules for two mass functions $m_1$ and $m_2$ on the recognition framework are:

$$m_1 \oplus m_2(a) = \frac{1}{K} \sum_{b \cap c = a} m_1(b) \cdot m_2(c) \tag{3}$$

where $m_1$ and $m_2$ represent two evidence chains, $a$, $b$, and $c$ denote the influencing factors of these chains, respectively, and K is the normalizing constant, which represents the conflict between evidences. The conflict between evidence elements is more severe when K is closer to 1, and the evidence sources are more consistent when K closer is to 0.

$$K = \sum_{b \cap c \neq \varnothing} m_1(b) \cdot m_2(c) = 1 - \sum_{b \cap c = \varnothing} m_1(b) \cdot m_2(c) \tag{4}$$

(2)     Rules for synthesizing multiple basic probability allocation functions

For $\forall a \subseteq M$, the Dempster synthesis rule for identifying a finite number of mass functions $m_1, m_2, m_3, \ldots, m_n$ on the framework is:

$$(m_1 \oplus m_2 \oplus \cdots \oplus m_n)(a) = \frac{1}{K} \sum_{a_1 \cap a_2 \cap \ldots \cap a_n = a} m_1(a_1) \cdot m_2(a_2) \cdots m_n(a_n) \tag{5}$$

where $m_1, m_2, m_3, \ldots, m_n$ represent $n$ evidence chains on the recognition framework, and $a_1, a_2, \ldots, a_n$ are the influencing factors of $n$ evidence chains.

$$\begin{aligned} K &= \sum_{a_1 \cap \ldots \cap a_n \neq \varnothing} m_1(a_1) \cdot m_2(a_2) \cdots m_n(a_n) \\ &= 1 - \sum_{a_1 \cap \ldots \cap a_n = \varnothing} m_1(a_1) \cdot m_2(a_2) \cdots m_n(a_n) \end{aligned} \tag{6}$$

*3.2. Improved DS Evidence Theory Method*
3.2.1. Method for Measuring the Degree of Conflict

The conventional DS evidence theory method states that if K is close to 1, evidence sources are in high conflict, and the results calculated using DS synthesis rules contradict objective facts. Therefore, numerous scholars have proposed various methods to measure evidence-source conflicts in response to this issue, with common methods tabulated in Table 1.

**Table 1.** Methods for measuring evidence-source conflicts.

| Method | Formula |
|---|---|
| Conflict factor K | $K_{1,2} = \sum\limits_{A_i \cap A_j = \varnothing} m_1(A_i) m_2\left(A_j\right)$ |
| Cosine value of the included angle [57] | $cor(m_1, m_2) = \dfrac{\langle m_1, m_2 \rangle}{|m_1| \cdot |m_2|}$ |
| Pignistic probability distance [58] | $difBetP_{m_1}^{m_2} = \dfrac{1}{2} \sum\limits_{a \subseteq M} \left(\left|BetP_{m_1}(a) - BetP_{m_2}(a)\right|\right)$ |
| Jousselme distance [59] | $d(m_1, m_2) = \sqrt{\dfrac{1}{2}(m_1 - m_2) D(m_1 - m_2)^{\mathrm{T}}}$ |
| BJS divergence [60] | $BJS(m_1, m_2) = \dfrac{\left[S\left(m_1, \frac{m_1 + m_2}{2}\right) + S\left(m_2, \frac{m_1 + m_2}{2}\right)\right]}{2}$ |
| Confidence Hellinger distance [61] | $D_H(m_1, m_2) = \dfrac{1}{\sqrt{2}} \left\| \sqrt{m_1(a_i)} - \sqrt{m_2(a_i)} \right\|_2$ |

**Example 1.** *Assuming a recognition framework* $M = \{\theta_1, \theta_2, \cdots, \theta_{20}\}$, *and two information sensors are converted into BPA functions as follows:*

① $\begin{cases} m_1(\theta_1) = 1 \\ m_2(\theta_{20}) = 1 \end{cases}$

② $\begin{cases} m_1(\theta_1) = 1/2, \ m_1(\theta_2) = 1/2 \\ m_2(\theta_{20}) = 1/2, \ m_2(\theta_{19}) = 1/2 \end{cases}$

③ $\begin{cases} m_1(\theta_1) = 1/3, \ m_1(\theta_2) = 1/3, \ m_1(\theta_3) = 1/3 \\ m_2(\theta_{20}) = 1/3, \ m_2(\theta_{19}) = 1/3, \ m_2(\theta_{18}) = 1/3 \end{cases}$

④ $\begin{cases} m_1(\theta_1) = 1/4, \ m_1(\theta_2) = 1/4, \ m_1(\theta_3) = 1/4, \ m_1(\theta_4) = 1/4 \\ m_2(\theta_{20}) = 1/4, \ m_2(\theta_{19}) = 1/4, \ m_2(\theta_{18}) = 1/4, \ m_2(\theta_{17}) = 1/4 \end{cases}$

⑤ $\begin{cases} m_1(\theta_1) = 1/5, \ m_1(\theta_2) = 1/5, \ m_1(\theta_3) = 1/5, \ m_1(\theta_4) = 1/5, \ m_1(\theta_5) = 1/5 \\ m_2(\theta_{20}) = 1/5, \ m_2(\theta_{19}) = 1/5, \ m_2(\theta_{18}) = 1/5, \ m_2(\theta_{17}) = 1/5, \ m_2(\theta_{16}) = 1/5 \end{cases}$

⑥ $\begin{cases} m_1(\theta_1) = 1/6, \ m_1(\theta_2) = 1/6, \ m_1(\theta_3) = 1/6, \ m_1(\theta_4) = 1/6, \ m_1(\theta_5) = 1/6, \ m_1(\theta_6) = 1/6 \\ m_2(\theta_{20}) = 1/6, \ m_2 m_2(\theta_{19}) = 1/6, \ m_2(\theta_{18}) = 1/6, \ m_2(\theta_{17}) = 1/6, \ m_2(\theta_{16}) = 1/6, \ m_2(\theta_{15}) = 1/6 \end{cases}$

⑦ $\begin{cases} m_1(\theta_1) = 1/7, \ m_1(\theta_2) = 1/7, \ m_1(\theta_3) = 1/7, \ m_1(\theta_4) = 1/7, \ m_1(\theta_5) = 1/7, \ m_1(\theta_6) = 1/7, \ m_1(\theta_7) = 1/7 \\ m_2(\theta_{20}) = 1/7, \ m_2(\theta_{19}) = 1/7, \ m_2(\theta_{18}) = 1/7, \ m_2(\theta_{17}) = 1/7, \ m_2(\theta_{16}) = 1/7, \ m_2(\theta_{15}) = 1/7, \ m_2(\theta_{14}) = 1/7 \end{cases}$

⑧ $\begin{cases} m_1(\theta_1) = 1/8, \ m_1(\theta_2) = 1/8, \ m_1(\theta_3) = 1/8, \ m_1(\theta_4) = 1/8, \ m_1(\theta_5) = 1/8, \ m_1(\theta_6) = 1/8, \ m_1(\theta_7) = 1/8, \ m_1(\theta_8) = 1/8 \\ m_2(\theta_{20}) = 1/8, \ m_2(\theta_{19}) = 1/8, \ m_2(\theta_{18}) = 1/8, \ m_2(\theta_{17}) = 1/8, \ m_2(\theta_{16}) = 1/8, \ m_2 m_2(\theta_{15}) = 1/8, \ m_2(\theta_{14}) = 1/8, \ m_2(\theta_{13}) = 1/8 \end{cases}$

⑨ $\begin{cases} m_1(\theta_1) = 1/9, \ m_1(\theta_2) = 1/9, \ m_1(\theta_3) = 1/9, \ m_1(\theta_4) = 1/9, \ m_1(\theta_5) = 1/9, \ m_1(\theta_6) = 1/9, \ m_1(\theta_7) = 1/9, \ m_1(\theta_8) = 1/9, \ m_1(\theta_9) = 1/9 \\ m_2(\theta_{20}) = 1/9, \ m_2(\theta_{19}) = 1/9, \ m_2(\theta_{18}) = 1/9, \ m_2(\theta_{17}) = 1/9, \ m_2(\theta_{16}) = 1/9, \ m_2(\theta_{15}) = 1/9, \ m_2(\theta_{14}) = 1/9, \ m_2(\theta_{13}) = 1/9, \ m_2(\theta_{12}) = 1/9 \end{cases}$

⑩ $\begin{cases} m_1(\theta_1) = 1/10, \ m_1(\theta_2) = 1/10, \ m_1(\theta_3) = 1/10, \ m_1(\theta_4) = 1/10, \ m_1(\theta_5) = 1/10, \ m_1(\theta_6) = 1/10, \ m_1(\theta_7) = 1/10, \ m_1(\theta_8) = 1/10, \ m_1(\theta_9) = 1/10, \ m_1(\theta_{10}) = 1/10 \\ m_2(\theta_{20}) = 1/10, \ m_2(\theta_{19}) = 1/10, \ m_2(\theta_{18}) = 1/10, \ m_2(\theta_{17}) = 1/10, \ m_2(\theta_{16}) = 1/10, \ m_2(\theta_{15}) = 1/10, \ m_2(\theta_{14}) = 1/10, \ m_2(\theta_{13}) = 1/10, \ m_2(\theta_{12}) = 1/10, \ m_2(\theta_{11}) = 1/10 \end{cases}$

From Example 1, it is apparent that $m_1$ and $m_2$ in Scenarios ① to ⑩ are all in complete conflict. For a specific recognition framework M, conflict factor K, cosine value of the included angle, Pignistic probability distance, Josselme distance, BJS divergence, and confidence Hellinger distance are employed to measure the conflict, as depicted in Table 2.

**Table 2.** Calculation results of Example 1.

| Conflict Measurement Method | Conflict Factor K | Cosine Value of the Included Angle | Pignistic Probability Distance | Jousselme Distance | BJS Divergence | Confidence Hellinger Distance |
|---|---|---|---|---|---|---|
| ① | 1 | 0 | 1 | 1 | 0.3010 | 1 |
| ② | 1 | 0 | 0.5000 | 0.7071 | 0.3010 | 1 |
| ③ | 1 | 0 | 0.3333 | 0.5774 | 0.3010 | 1 |
| ④ | 1 | 0 | 0.2500 | 0.5000 | 0.3010 | 1 |
| ⑤ | 1 | 0 | 0.2000 | 0.4472 | 0.3010 | 1 |
| ⑥ | 1 | 0 | 0.1667 | 0.4082 | 0.3010 | 1 |
| ⑦ | 1 | 0 | 0.1429 | 0.3780 | 0.3010 | 1 |
| ⑧ | 1 | 0 | 0.1250 | 0.3536 | 0.3010 | 1 |
| ⑨ | 1 | 0 | 0.1111 | 0.3333 | 0.3010 | 1 |
| ⑩ | 1 | 0 | 0.1000 | 0.3162 | 0.3010 | 1 |

Considering the complete conflict between two sets of BPA functions in Scenarios ① to ⑩ in Example 1, the conflict factors of $m_1$ and $m_2$ should reach the maximum value of 1. From Table 2, it is obvious that as the number of propositional elements increases, the conflict factors obtained using the Jousselme distance and the Pignistic probability distance decrease, contradicting objective facts. As the number of propositional elements expands, the conflict measurement results calculated using BJS divergence remain constant. Nonetheless, in Example 1, all 10 cases' $m_1$ and $m_2$ functions are in complete conflict, and the calculated conflict measurement value should be 1. Hence, the conflict measurement results obtained using BJS divergence are not consistent with the facts. The conflict measurement results obtained using the cosine value of the included angle are all 0, suggesting no conflict at all in all 10 cases of Example 1, which contradicts objective facts. Comparing the above six methods, using the confidence Hellinger distance and conflict factor K can effectively measure the degree of conflict between two sets of BPA functions in the case of complete conflict.

**Example 2.** *Assumes the BPA of recognition framework $M = \{a, b, c\}$, evidence chain $m_1$, and evidence chain $m_2$ are as follows:*

$$m_1 : m_1(a) = 0.4, m_1(b) = 0.4, m_1(c) = 0.2$$
$$m_2 : m_2(a) = 0.4, m_2(b) = 0.4, m_2(c) = 0.2$$

Using two methods, the confidence Hellinger distance and conflict factor K, to calculate the conflict measures of the above two pieces of evidence, we obtain: $D_H(m_1, m_2) = 0$, Conflict factor K = $1 - 0.4 \times 0.4 \times 2 - 0.2 \times 0.2 = 0.64$. From Example 2, it is apparent that the evidence chains $m_1$ and $m_2$ are not in conflict at all. The conflict measure, determined using the confidence Hellinger distance, yields a value of 0, which aligns with objective facts. The conflict measure calculated using the conflict factor K is 0.64, demonstrating a significant conflict between two evidence chains, which contradicts objective facts. To sum up, it appears that the confidence Hellinger distance can more accurately reflect the conflict measurement values between two evidence chains.

### 3.2.2. Evidence Weight Acquisition

Suppose $m_a$ and $m_b$ (*a*, *b* = 1, 2, …, *n*) are two sets of BPA functions under the identification framework. Initially, we calculate the conflict measurement values between

two evidence chains, $m_a$, and $m_b$, using the confidence Hellinger distance. The similarity between these evidence chains is expressed as follows:

$$\text{Sim}_{ab} = 1 - D_H(m_a, m_b) \tag{7}$$

Based on this evidence similarity, the evidence support can be computed. The formula for calculating the evidence support is as follows:

$$\text{Sup}_a = \sum_{b,a \neq b}^{n} \text{Sim}_{ab} \tag{8}$$

It is generally accepted that the credibility of an evidence chain increases with the support it receives. The formula for calculating the weight of the evidence chain is given by:

$$\omega_a = \frac{\text{Sup}_a}{\sum_{b=1}^{n} \text{Sup}_b} \tag{9}$$

### 3.2.3. Conflict Evidence Chain Correction

The traditional DS evidence theory suggests that if the conflict factor K of two evidence chains equals 1, it implies a complete conflict between these chains, rendering the DS evidence theory invalid. To fully utilize the effective information of the original project monitoring data, conflict-free monitoring data will remain uncorrected. Only monitoring data with higher conflict measurement values will undergo correction, and the corrected weighted average evidence will replace the original monitoring data. We will fuse the corrected evidence using the DS evidence theory method to yield the fusion results of multi-source heterogeneous data.

The formula for calculating the weighted average evidence chain is:

$$m_a'(A) = \sum_{a=1}^{n} \omega_a \times m_a(A) \tag{10}$$

## 4. Safety Risk Assessment of Nearshore Tunnel Construction Based on the Improved DS Evidence Theory

### 4.1. Project Overview

A specific nearshore tunnel extends from stake number K6+040 to stake number K8+320, covering a total length of 2280 m. The section layout includes 1130 m (U-shaped groove) + (45 + 2560 + 45) m (hidden box section) + 1230 m (U-shaped groove). Depending on different structural forms, the tunnel's primary structure is split into five parts: the north open section, the north light transition section, the open excavation and buried section, the south light transition section, and the south open section. The specific section mileage is detailed in Table 3.

**Table 3.** Scale table of a certain nearshore tunnel.

| Structural Form | Starting and Ending Mileage | Paragraph Length (m) |
|---|---|---|
| North open section | K6+040–K6+295 | 255 |
| North light transition section | K6+295–K6+370 | 75 |
| Open excavation and buried section | K6+370–K7+960 | 1590 |
| South light transition section | K7+960–K8+035 | 75 |
| South open section | K8+035–K8+320 | 285 |
| Tunnel building length | | 2280 |

Following the recommendations of the "Discussion Meeting on Monitoring and Measurement Schemes for a Certain Nearshore Tunnel Project in Ningbo", monitoring shall be conducted on the second level foundation pit for excavation depths below 5 m, while the

first level foundation pit monitoring applies to depths exceeding 5 m in the middle. Therefore, the first level foundation pit standard monitoring range includes K6+415–K6+955 and K7+435–K7+915, with the remaining sections monitored according to the second level foundation pit standard. This study's selected monitoring data belongs to K7+435–K7+915, hence monitoring is performed as per the first level foundation pit. The project's main monitoring content is outlined in Table 4.

**Table 4.** Project monitoring content.

| Measuring Item | Device Name | Equipment Model | Monitoring Accuracy |
|---|---|---|---|
| Groundwater level | Steel ruler water level gauge | DiNi03 SWJ-8090 | ±0.3 mm/km ±1 mm |
| Pore water pressure | Vibrating wire frequency reading instrument | 609 | ±0.1 Hz |
| Vertical displacement of the top of the retaining pile | Tianbao Dumpy level | Trimble DiNi03 | ±0.3 mm/km |
| Soil pressure | Vibrating wire frequency reading instrument | 609 | ±0.1 Hz |
| Support axial force | Vibrating wire frequency reading instrument | 609 | ±0.1 Hz |
| Horizontal displacement of the top of the retaining pile | Vibrating wire frequency reading instrument | TS09 plus 1″R500 | 1.5 + 2 ppm |

*4.2. Monitoring Data*

4.2.1. Data Summary

To manage abnormal conditions such as displacement and settlement of the enclosure structure, surrounding buildings, structures, and underground pipelines, and to decrease safety risk probability during the construction process, a rigorous monitoring network was established throughout the construction process of the Ningbo nearshore tunnel, aiming for information-based construction. This article evaluated the data from five monitoring points, including K7+880–K7+800, of the Ningbo nearshore tunnel. The monitoring data are displayed in Tables 5 and 6.

**Table 5.** Summary of vertical displacement, horizontal displacement, and groundwater level monitoring data at the top of the retaining pile.

| Monitoring Point | Vertical Displacement of the Top of the Retaining Pile (mm) | Horizontal Displacement of the Top of the Retaining Pile (mm) | Groundwater Level (mm) |
|---|---|---|---|
| K7+880 | −1.10 | 16.33 | 845 |
| K7+860 | 3.25 | 13.80 | 372 |
| K7+840 | 34.46 | 14.50 | 1997 |
| K7+820 | −1.30 | 4.90 | 523 |
| K7+800 | 1.35 | 6.25 | 1409 |

**Table 6.** Summary of monitoring data for soil pressure, support axial force, and pore water pressure.

| Monitoring Point | Soil Pressure (kPa) | Support Axial Force (kN) | Pore Water Pressure (kPa) |
|---|---|---|---|
| K7+880 | 11.12 | −277.89 | 18.56 |
| K7+860 | 7.01 | 163.84 | 8.55 |
| K7+840 | 33.61 | −1411.66 | −0.92 |
| K7+820 | −20.83 | 437.17 | 4.79 |
| K7+800 | −28.65 | 220.69 | −32.98 |

The monitoring data for vertical displacement of the top of the retaining pile, horizontal displacement of the retaining pile, soil pressure, pore water pressure, support axial force, and groundwater level at each monitoring point are illustrated in Figures 1–6. For

simplification and clarity within these graphs, the naming conventions for the monitoring points have been altered. For instance, monitoring point K7+880 is abbreviated to 7880 in the graphical representation. Relative $Y$-axis labels are used to display each point.

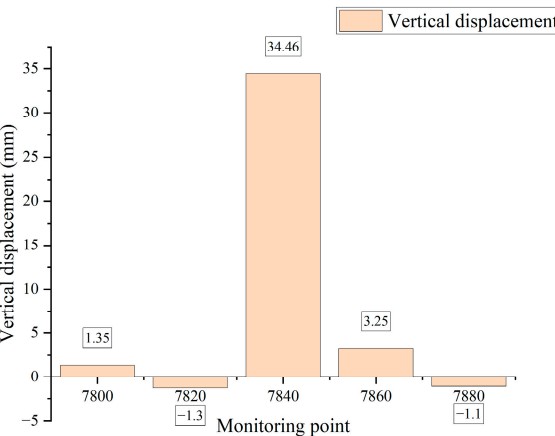

**Figure 1.** Vertical displacement at each monitoring point.

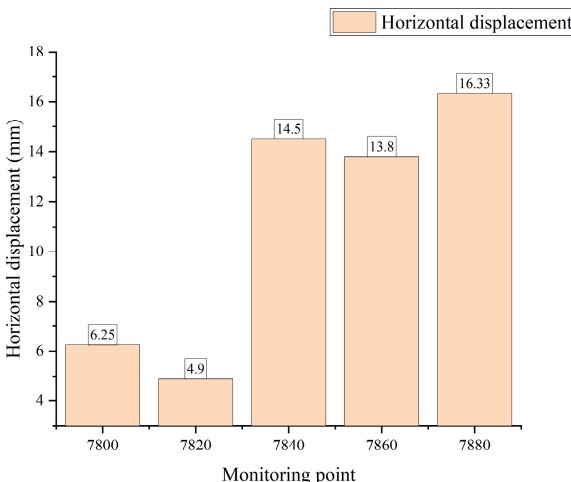

**Figure 2.** Horizontal displacement at each monitoring point.

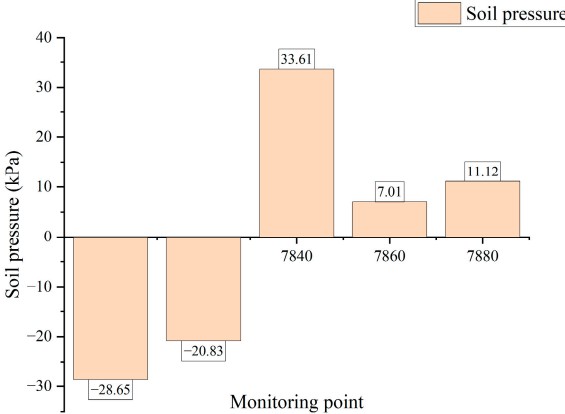

**Figure 3.** Soil pressure at each monitoring point.

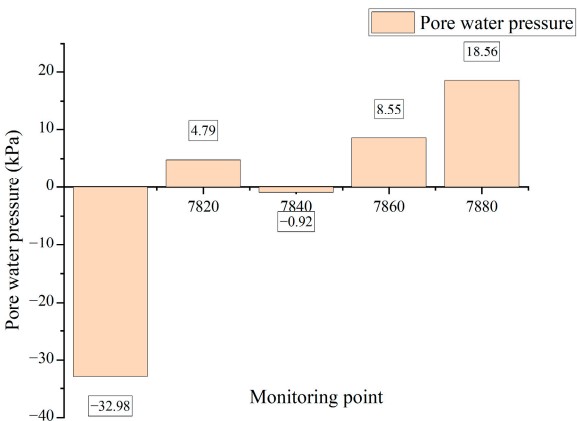

**Figure 4.** Pore water pressure at each monitoring point.

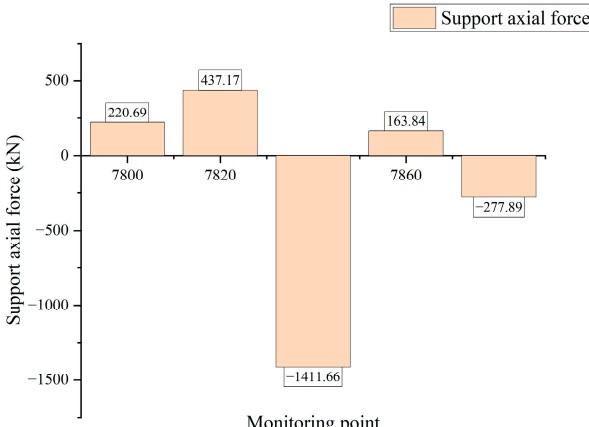

**Figure 5.** Support axial force at each monitoring point.

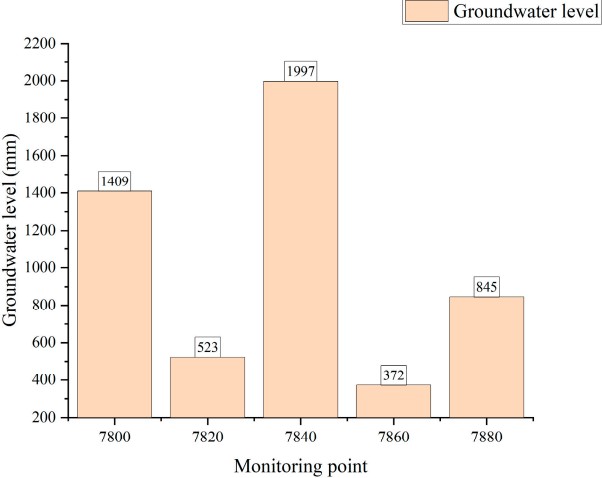

**Figure 6.** Groundwater level at each monitoring point.

### 4.2.2. Index Grading

This experiment is oriented to the example of the nearshore tunnel-specific indicators of evaluation level classification standards, strictly adhering to the guidelines set forth in the Chinese standard [62], "Technical Code for Monitoring of Building Foundation Pit Engineering" (GB50497-2009). Risk levels at the monitoring points are categorized into three distinct levels:

Level I risk indicates that the safety conditions at the monitoring point are well within controlled limits. When the data fusion results show that the measurement point is Level I risk, it suggests that the risk of the measurement point is manageable and under control.

Level II risk indicates that the risk of the monitoring point is moderate, and it is necessary to investigate factors compromising construction safety. This entails an increase in the frequency of monitoring and implementation of tailored safety technical measures, followed by an audit. When the data fusion results show that the measurement point is Level II risk, it is imperative to formulate corresponding countermeasures based on the actual construction site conditions.

Level III risk indicates that the risk of the monitoring point is severe, necessitating the development of a specialized construction plan and emergency response measures. Enhanced monitoring frequency is required, and localized or segmented warning technologies should be employed. When the data fusion results show that the measurement point is Level III risk, it implies that the measurement point is significantly dangerous, and the construction should be terminated immediately and the actual situation at the construction site should be considered.

The specific risk indicators are presented in Table 7.

**Table 7.** Risk indicator grading.

| Serial Number | Monitoring Item | Safety Level of the Foundation Pit Design | | |
| --- | --- | --- | --- | --- |
| | | I | II | III |
| 1 | Vertical displacement of the top of the retaining pile | ±7 mm | ±8.5 mm | ±10 mm |
| 2 | Horizontal displacement of the top of the retaining pile | ±17.5 mm | ±21.25 mm | ±25 mm |
| 3 | Soil pressure | 60%–70%$f_1$ | 70%–80%$f_1$ | 70%–80%$f_1$ |
| 4 | Pore water pressure | 60%–70%$f_1$ | 70%–80%$f_1$ | 70%–80%$f_1$ |
| 5 | Groundwater level | ±1000 mm | ±1500 mm | ±2000 mm |
| 6 | Support axial force | 966 kN | 1173 kN | 1380 kN |

Note: $f_1$—Design value of the load.

*4.3. Construction Safety Risk Assessment*

4.3.1. Normalization of Indicators

Monitoring data indicators are normalized to make them dimensionless, thereby enhancing their comparability. If monitoring indicators negatively affect construction safety risks, normalization is shown in Formula (11).

$$m'_1 = (m_{1max} - m_1)/(m_{1max} - m_{1min}) \tag{11}$$

On the other hand, if the monitoring indicators positively affect construction safety risks, normalization is shown in Formula (12).

$$m'_1 = (m_1 - m_{1min})/(m_{1max} - m_{1min}) \tag{12}$$

Key monitoring indicators for a specific nearshore tunnel construction in Ningbo have been normalized, with the specific data displayed in Table 8. The BPA functions $m_i$ and $m_j (i, j = 1, 2, 3, 4, 5, 6)$ on a certain recognition framework are established to monitor the vertical and horizontal displacements of the top of the retaining pile, soil pressure, pore water pressure, groundwater level, and support axial force, $M = \{a, b, c, d\}$.

**Table 8.** Normalization processing.

| Monitoring Point | Monitoring Item | Risk Level | | | | Actual Data |
|---|---|---|---|---|---|---|
| | | Normal(*a*) | I(*b*) | II(*c*) | III(*d*) | |
| K7+880 | Vertical displacement of the top of the retaining pile | 0–0.55 | 0.55–0.7 | 0.7–0.85 | 0.85–1 | 0.20 |
| | Horizontal displacement of the top of the retaining pile | 0–0.55 | 0.55–0.7 | 0.7–0.85 | 0.85–1 | 0.58 |
| | Groundwater level | 0–0.50 | 0.50–0.63 | 0.63–0.75 | 0.75–1 | 0.19 |
| | Pore water pressure | 0–0.75 | 0.75–0.83 | 0.83–0.91 | 0.91–1 | 0.06 |
| | Soil pressure | 0–0.75 | 0.75–0.83 | 0.83–0.91 | 0.91–1 | 0.11 |
| | Support axial force | 0–0.55 | 0.55–0.7 | 0.7–0.85 | 0.85–1 | 0.02 |
| K7+860 | Vertical displacement of the top of the retaining pile | 0–0.55 | 0.55–0.7 | 0.7–0.85 | 0.85–1 | 0.15 |
| | Horizontal displacement of the top of the retaining pile | 0–0.55 | 0.55–0.7 | 0.7–0.85 | 0.85–1 | 0.55 |
| | Groundwater level | 0–0.50 | 0.50–0.63 | 0.63–0.75 | 0.75–1 | 1.00 |
| | Pore water pressure | 0–0.75 | 0.75–0.83 | 0.83–0.91 | 0.91–1 | 0.06 |
| | Soil pressure | 0–0.75 | 0.75–0.83 | 0.83–0.91 | 0.91–1 | 0.11 |
| | Support axial force | 0–0.55 | 0.55–0.7 | 0.7–0.85 | 0.85–1 | 0.02 |
| K7+840 | Vertical displacement of the top of the retaining pile | 0–0.55 | 0.55–0.7 | 0.7–0.85 | 0.85–1 | 0.02 |
| | Horizontal displacement of the top of the retaining pile | 0–0.55 | 0.55–0.7 | 0.7–0.85 | 0.85–1 | 0.58 |
| | Groundwater level | 0–0.50 | 0.50–0.63 | 0.63–0.75 | 0.75–1 | 0.20 |
| | Pore water pressure | 0–0.75 | 0.75–0.83 | 0.83–0.91 | 0.91–1 | 0.06 |
| | Soil pressure | 0–0.75 | 0.75–0.83 | 0.83–0.91 | 0.91–1 | 0.11 |
| | Support axial force | 0–0.55 | 0.55–0.7 | 0.7–0.85 | 0.85–1 | 0.02 |
| K7+820 | Vertical displacement of the top of the retaining pile | 0–0.55 | 0.55–0.7 | 0.7–0.85 | 0.85–1 | 0.20 |
| | Horizontal displacement of the top of the retaining pile | 0–0.55 | 0.55–0.7 | 0.7–0.85 | 0.85–1 | 0.19 |
| | Groundwater level | 0–0.50 | 0.50–0.63 | 0.63–0.75 | 0.75–1 | 0.59 |
| | Pore water pressure | 0–0.75 | 0.75–0.83 | 0.83–0.91 | 0.91–1 | 0.06 |
| | Soil pressure | 0–0.75 | 0.75–0.83 | 0.83–0.91 | 0.91–1 | 0.11 |
| | Support axial force | 0–0.55 | 0.55–0.7 | 0.7–0.85 | 0.85–1 | 0.02 |
| K7+800 | Vertical displacement of the top of the retaining pile | 0–0.55 | 0.55–0.7 | 0.7–0.85 | 0.85–1 | 0.26 |
| | Horizontal displacement of the top of the retaining pile | 0–0.55 | 0.55–0.7 | 0.7–0.85 | 0.85–1 | 0.10 |
| | Groundwater level | 0–0.50 | 0.50–0.63 | 0.63–0.75 | 0.75–1 | 0.36 |
| | Pore water pressure | 0–0.75 | 0.75–0.83 | 0.83–0.91 | 0.91–1 | 0.05 |
| | Soil pressure | 0–0.75 | 0.75–0.83 | 0.83–0.91 | 0.91–1 | 0.11 |
| | Support axial force | 0–0.55 | 0.55–0.7 | 0.7–0.85 | 0.85–1 | 0.02 |

### 4.3.2. Obtaining Index Weights

We calculate the degree of conflict $D_H(m_i, m_j)$ between normalized BPA functions $m_i$ and $m_j (i, j = 1, 2, \ldots, 6)$ using the confidence Hellinger distance, forming a conflict matrix $D_H$, as expressed in Formula (13).

$$D_H = \begin{bmatrix} 0 & \cdots & D_H(m_1, m_j) & \cdots & D_H(m_1, m_n) \\ \vdots & & \vdots & & \vdots \\ D_H(m_i, m_1) & \cdots & D_H(m_i, m_j) & \cdots & D_H(m_i, m_n) \\ \vdots & & \vdots & & \vdots \\ D_H(m_n, m_1) & \cdots & D_H(m_n, m_j) & \cdots & D_H(m_n, m_n) \end{bmatrix} \quad (13)$$

Based on Formula (13), we calculate the matrix and obtain:

$$D_H(m_i, m_j)(i, j = 1, 2, \ldots, 6) = \begin{bmatrix} 0 & 0 & 0.090 & 0.149 & 0.149 & 0 \\ 0 & 0 & 0.090 & 0.149 & 0.149 & 0 \\ 0.090 & 0.090 & 0 & 0.194 & 0.194 & 0.090 \\ 0.149 & 0.149 & 0.194 & 0 & 0 & 0.149 \\ 0.149 & 0.149 & 0.194 & 0 & 0 & 0.149 \\ 0 & 0 & 0.090 & 0.149 & 0.149 & 0 \end{bmatrix}$$

We then calculate the similarity between two evidence chains $m_i$ and $m_j$ according to Formula (7), and obtain:

$$Sim(m_i, m_j)(i, j = 1, 2, \ldots, 6) = \begin{bmatrix} 1 & 1 & 0.910 & 0.851 & 0.851 & 1 \\ 1 & 1 & 0.910 & 0.851 & 0.851 & 1 \\ 0.910 & 0.910 & 1 & 0.806 & 0.806 & 0.910 \\ 0.851 & 0.851 & 0.806 & 1 & 1 & 0.851 \\ 0.851 & 0.851 & 0.806 & 1 & 1 & 0.851 \\ 1 & 1 & 0.910 & 0.851 & 0.851 & 1 \end{bmatrix}$$

Next, we calculate the support of the evidence chain according to Formula (8), and obtain:

$$Sup_1 = \sum_{j, j \neq 1}^{6} Sim_{1j} = 4.612, Sup_2 = \sum_{j, j \neq 2}^{6} Sim_{2j} = 4.612, Sup_3 = \sum_{j, j \neq 3}^{6} Sim_{3j} = 4.342,$$
$$Sup_4 = \sum_{j, j \neq 4}^{6} Sim_{4j} = 4.359, Sup_5 = \sum_{j, j \neq 5}^{6} Sim_{5j} = 4.359, Sup_6 = \sum_{j, j \neq 6}^{6} Sim_{6j} = 4.612$$

Finally, we calculate the weight of the evidence chain according to Formula (9), and obtain:

$$\omega_1 = \frac{Sup_1}{\sum_{j=1}^{j=6} Sup_j} = 0.171, \omega_2 = \frac{Sup_2}{\sum_{j=1}^{j=6} Sup_j} = 0.171, \omega_3 = \frac{Sup_3}{\sum_{j=1}^{j=6} Sup_j} = 0.161,$$
$$\omega_4 = \frac{Sup_4}{\sum_{j=1}^{j=6} Sup_j} = 0.162, \omega_5 = \frac{Sup_5}{\sum_{j=1}^{j=6} Sup_j} = 0.162, \omega_6 = \frac{Sup_6}{\sum_{j=1}^{j=6} Sup_j} = 0.171$$

### 4.3.3. Indicator Fusion

We correct the conflict indicators based on Formula (10) and fuse the corrected indicators according to Formula (5). The fusion results of indicators are displayed in Table 9.

**Table 9.** Indicator fusion results.

| Monitoring Point | Risk Level | | | | Fusion Results | Risk Level |
| --- | --- | --- | --- | --- | --- | --- |
| | Normal | I | II | III | | |
| K7+880 | 0–0.61 | 0.61–0.73 | 0.73–0.85 | 0.85–1 | 0.58 | Normal |
| K7+860 | 0–0.61 | 0.61–0.73 | 0.73–0.85 | 0.85–1 | 0.78 | II |
| K7+840 | 0–0.61 | 0.61–0.73 | 0.73–0.85 | 0.85–1 | 0.58 | Normal |
| K7+820 | 0–0.61 | 0.61–0.73 | 0.73–0.85 | 0.85–1 | 0.59 | Normal |
| K7+800 | 0–0.61 | 0.61–0.73 | 0.73–0.85 | 0.85–1 | 0.36 | Normal |

Using the improved DS evidence theory method, construction safety risk assessment was performed on five monitoring points of a nearshore tunnel in Ningbo. The results classify monitoring point K7+860 as a Level II risk (as plotted in Figure 7), while the remaining monitoring points are in a normal state.

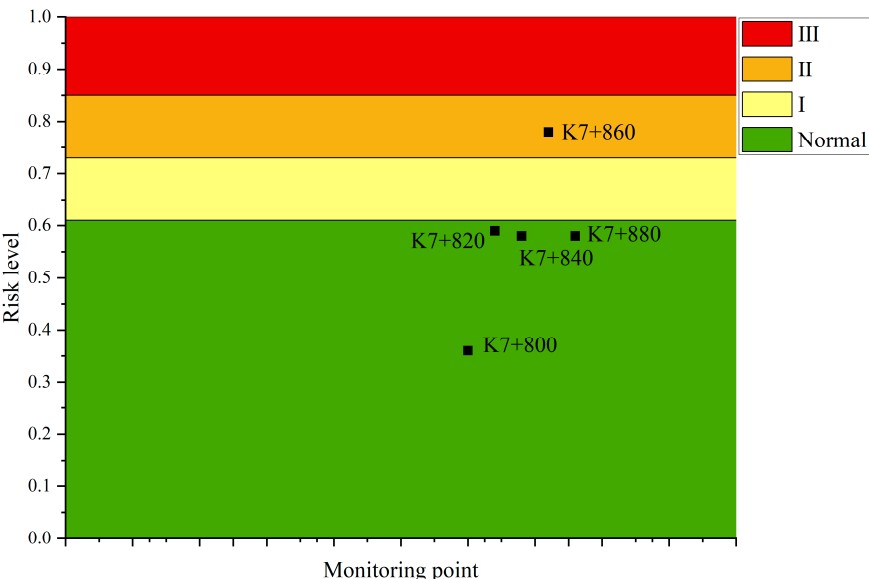

**Figure 7.** Distribution of data fusion results.

## 5. Safety Risk Warning for Nearshore Tunnel Construction Based on BIM Technology

*5.1. Revit Secondary Development Technology*

We create a new Visual Studio (VS) project and enter the vs. interface. Adding references to Windows, controls, and their namespaces, RevitAPI.dll, and RevitAPIUI.dll enables the calling of the IExternalCommand and IExternalApplication interfaces. By writing plugins through interfaces, generating .dll files, and concurrently writing .add files, we can activate or deactivate plugin functionalities. The secondary development process of Revit is illustrated in Figure 8.

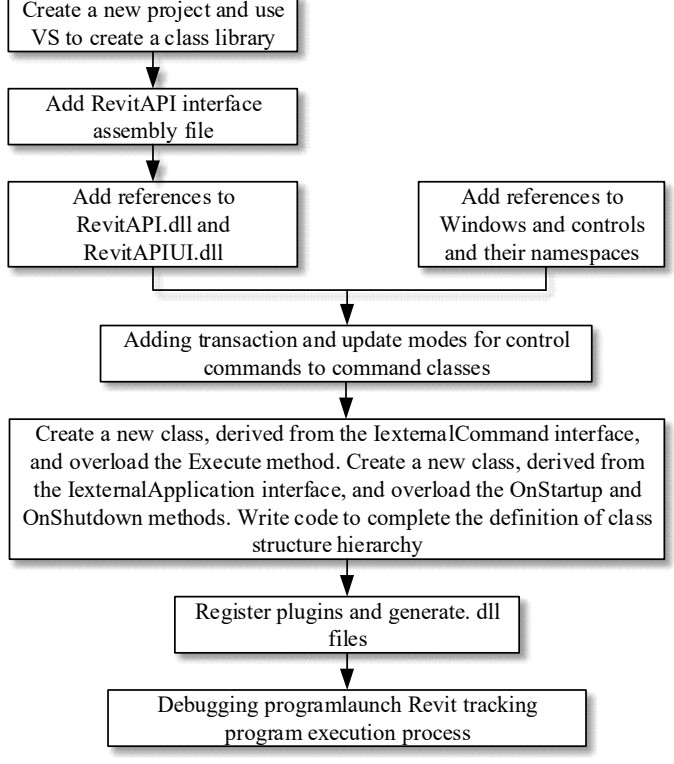

**Figure 8.** Revit secondary development process.

*5.2. Example-Visualization of Safety Risk Warning for Nearshore Tunnel Construction*

Currently, real-time construction monitoring information predominantly employs a file management approach. This method stores an extensive array of monitoring information, accrued through repeated observations over extended periods and across diverse metrics. Such an approach is susceptible to complications, including data structure disarray and file loss. Moreover, monitoring information is primarily displayed through graphs, which lack sufficient intuitiveness to quickly discern changes at specific points within a construction site.

To tackle these issues, Autodesk Revit serves as the selected platform for the visualization of construction monitoring information. By integrating the monitoring information into a centralized database and leveraging secondary development technology, Revit allows for the seamless import of monitoring point data into the BIM framework. This visual integration fosters an intuitive and convenient environment for stakeholders to assess the condition of each monitoring point. As a result, timely and appropriate countermeasures can be enacted for high-risk monitoring points, thereby reducing the likelihood of construction-related hazards. And the comparison between traditional construction monitoring and construction monitoring information visualization platform based on BIM is shown in Table 10.

**Table 10.** Comparison between traditional construction monitoring and construction monitoring information visualization platform based on BIM.

| | Traditional Construction Monitoring | Construction Monitoring Information Visualization Platform Based on BIM |
| --- | --- | --- |
| Data storage | Files (easily lead to disorganized data structures and file loss) | Database |
| Data updating | Manual updating and integration | System automatically updates and provides feedback on model changes |
| Data presentation | Non-intuitive charts and text | Presented graphically for an intuitive view of the status of each monitoring point, allowing for a rapid identification of risk locations through the model |
| Collaboration | Manual information transmission and coordination, which can result in information loss and misunderstanding) | Information sharing, enabling multi-party collaboration and communication to enhance management efficiency |

Initially, Revit software is employed to construct the BIM model of the tunnel's main structure. Figure 9a,b depict the 3D drawings, plan views, and cross-sectional views of this model.

Furthermore, in Figure 10, the layout of monitoring points K7+800–K7+880 within the BIM model of this nearshore tunnel is displayed.

Figure 11 presents the user interface (UI) for each monitoring point. Using point K7+860 as an example, after selecting the appropriate monitoring time, double-click "K7+860" to enter the monitoring point data input interface. This interface encompasses various data entry fields: monitoring date, monitoring point location, vertical displacement, horizontal displacement, soil pressure, pore water pressure, support axial force, and groundwater level. After inputting the relevant monitoring data, a double-click triggers a data fusion process that amalgamates these disparate monitoring indicators for comprehensive evaluation.

The backend system employs the improved DS evidence theory to automatically fuse the monitoring point data, and the fusion result's risk level corresponds to the monitoring equipment's warning level. For example, the risk assessment result of monitoring point K7+860 is Level II, and the corresponding monitoring equipment warning level is also Level II, as demonstrated in Figure 12.

**(a)**

**(b)**

**Figure 9.** BIM model view of the main structure of a submarine tunnel in Ningbo (China). (**a**) 3D BIM model drawing of the main structure of a submarine tunnel. (**b**) Cross-sectional view of the tunnel model.

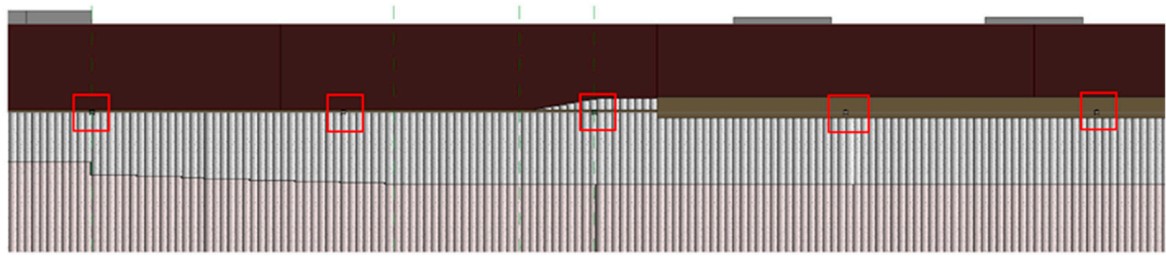

**Figure 10.** Layout of monitoring points K7+800–K7+880.

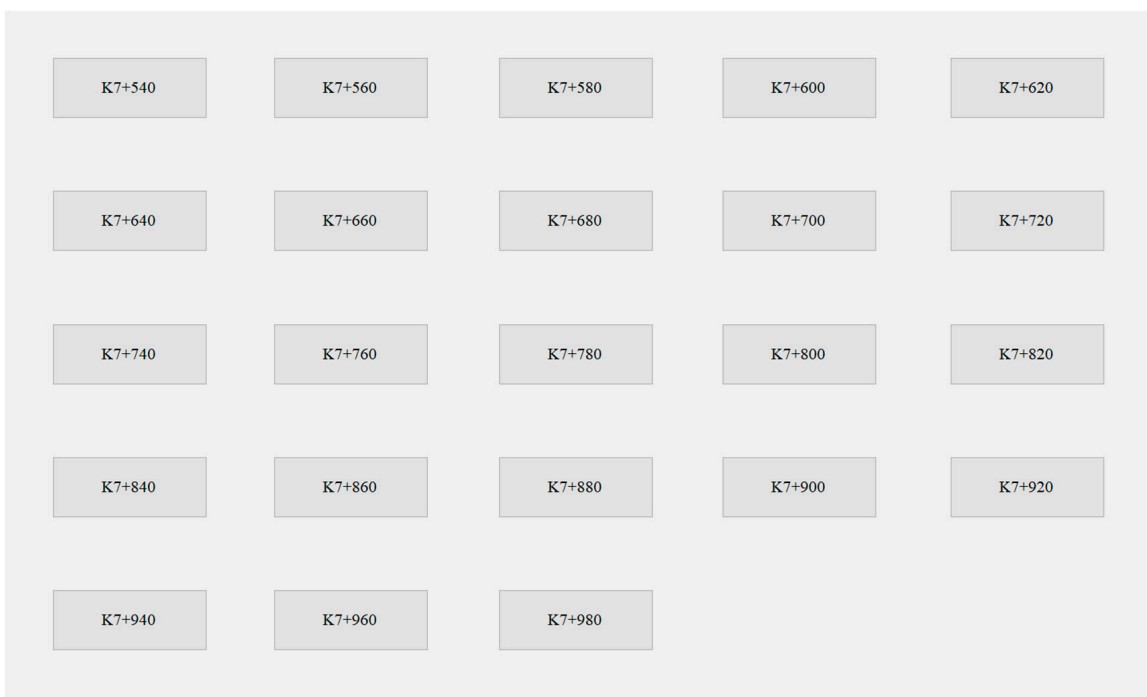

**Figure 11.** UI for each monitoring point.

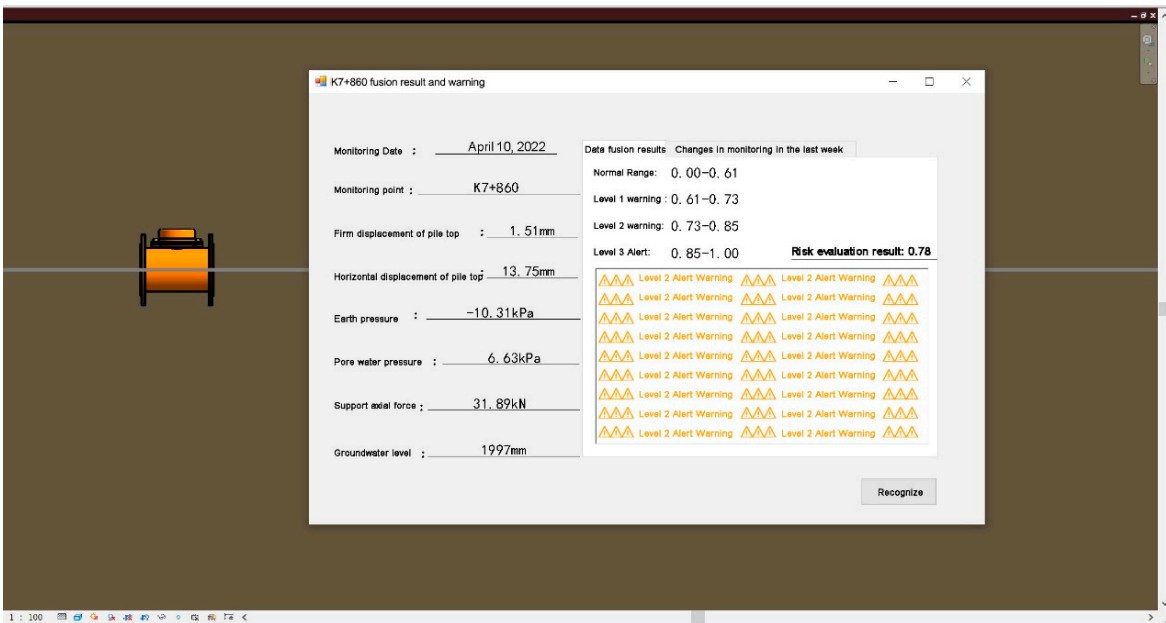

**Figure 12.** K7+860 monitoring point risk warning.

## 6. Discussion

In today's increasingly specialized, mechanized, and integrated construction sector, the critical issue of how to enhance construction precision and avoid recurrent safety incidents urgently demands resolution. Risk and hazard monitoring and assessment have always been at the forefront of the construction industry. Relying solely on manual labor for construction safety risk evaluation and management proves costly, inefficient, and is susceptible to errors or oversights. BIM technology, renowned for its digital and visual capabilities, has been extensively deployed across the entire life cycle of design, construction, operation, and other stages in recent years. Nevertheless, its application in construction monitoring and early warning remains under-researched.

To further enhance safety management during construction and increase efficiency, this paper proposes an intelligent evaluation and real-time warning system of safety risks in a nearshore tunnel construction based on BIM, considering the current application status in practical engineering. Taking a specific underwater tunnel in Ningbo as an example, we verified the feasibility and applicability of the evaluation method and software. The primary research outcomes of this article are as follows:

- Assuming the comparison of six methods for measuring the degree of conflict, namely, conflict factor K, cosine value of the included angle, Pignistic probability distance, Josselme distance, BJS divergence, and confidence Hellinger distance, it has been concluded that the confidence Hellinger distance provides the most accurate reflection of the conflict measurement value between two evidence chains. Through calculations involving the degree of conflict, credibility, and evidence weight, the conflict monitoring data chain undergoes modification. Subsequently, the modified evidence chain is fused using the Dempster rule. When a high conflict monitoring data chain is present, the improved DS evidence theory method can reasonably allocate conflicts, overcoming traditional method limitations and achieving more accurate fusion results.

- Using the data from five monitoring points of a nearshore tunnel in Ningbo as an example, we conduct a dynamic safety risk assessment of construction on the monitoring data. The results demonstrate that monitoring point K7+860 has a risk assessment level of II, while the other monitoring points are in a normal state. The dynamic safety risk level of monitoring points, derived from construction monitoring data through data fusion technology, offers a better reflection of the construction site's safety risk level compared to monitoring scattered data. Based on the safety risk assessment level of monitoring point K7+860, the data fusion-based safety risk assessment process is embedded into a computer through the C sharp programming language to achieve automatic monitoring data fusion functionality. The backend transmits the fusion results to the sensors in the BIM model. These sensors, in turn, utilize the fusion results to determine the dynamic safety risk level of the respective monitoring points. If the risk level exceeds the warning limit, the system will automatically sound an alarm and guide engineering personnel to address the situation.

## 7. Conclusions

### 7.1. Applications and Innovation

- Given the limitations of traditional construction safety risk assessment methodologies—namely, the influence of expert subjectivity and the underutilization of real-time construction site data—this article introduces a safety risk evaluation method for the construction of submarine tunnels based on the DS evidence theory. Utilizing construction monitoring data, this approach employs data fusion technology to dynamically determine the safety risk levels of various monitoring points. This fused data offers a more accurate reflection of site-wide safety risk compared to isolated, decentralized monitoring data.

- Furthermore, BIM technology's role in construction safety risk management has been explored. While BIM technology is widely adopted in the design, construction, and operational phases, its incorporation into construction monitoring and early warning systems remains underexplored. Therefore, this paper advances a real-time monitoring and early warning visualization framework for construction safety risks based on BIM technology. This innovation enhances the intuitiveness of risk warnings and mitigates the lag in early warnings, thereby reducing the likelihood of safety incidents during construction. This research supports the application of monitoring information in BIM frameworks, offering a more comprehensive basis for construction safety management for all project participants.

*7.2. Limitations and Future Work*

- This study contributes a novel methodology for evaluating safety risks in engineering construction, foundational theory for the integrated application of BIM technology and real-time construction site monitoring, as well as robust techniques for processing monitoring information. Despite these initial accomplishments, certain areas warrant further exploration, detailed as follows:
- The present research is chiefly concerned with construction risk management in nearshore tunnel environments. However, the methodologies employed hold broader application potential, including in areas like scaffold construction risk management. Scaffolding is ubiquitously utilized in construction activities and presents its own distinct set of risks—such as scaffold instability and material inadequacy. Applying the risk management approaches delineated in this study to scaffolding could substantially improve construction safety and operational efficiency.
- Given the extended observational periods and brief intervals intrinsic to construction projects, real-time monitoring is a critical element in effective engineering management. Future work should explore the deployment of automated sensor-based monitoring systems to facilitate real-time data collection and dynamic monitoring. Such technological advancements would bridge the existing gap between data acquisition and model development, thereby bolstering our capacity to monitor and issue timely alerts for, emerging risks.
- The data fusion techniques currently employed are optimized for structured data sets. An essential avenue for future research will be the expansion of these techniques to accommodate unstructured data, which can take various forms—including text, images, and audio recordings. Unstructured data carries considerable relevance for risk management applications. Merging this form of data with structured data will allow for more precise and accurate risk assessments, thereby enhancing the overall efficacy of risk management strategies.

**Author Contributions:** P.W.: Conceptualization, Methodology, Investigation, Writing—review & editing, Project administration, Funding acquisition. L.Y.: Writing—original draft, Visualization, Investigation, Formal analysis, Data curation. W.L.: Writing—review & editing, Investigation. J.H.: Investigation. Y.X.: Resources, Supervision. All authors have read and agreed to the published version of the manuscript.

**Funding:** This research was funded by the [Science and Technology Project of Ningbo Transportation Bureau] grant number [No. 202007], and [Science and Technology Project of Zhejiang Provincial Department of Transport] grant number [No. 202225].

**Institutional Review Board Statement:** Not applicable.

**Informed Consent Statement:** This study didn't involve humans.

**Data Availability Statement:** The data is unavailable due to privacy.

**Conflicts of Interest:** The authors declare that they have no known competing financial interests or personal relationships that could have appeared to influence the work reported in this paper.

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
