# Peer review of "Construction Safety Risk Assessment and Early Warning of Nearshore Tunnel Based on BIM Technology"

_jmse, doi:10.3390/jmse11101996_

Round 1
Reviewer 1 Report
The topic is of sufficient interest and its structure and organization are overall acceptable but can be improved. The manuscript’s innovation is debatable to some degree and needs to be further elaborated and justified. Moreover, robust verification/validation is required to enhance the quality of this manuscript. The following comments explain the manuscript’s drawbacks in detail and help improve its quality if addressed meticulously by the authors:
The authors may need to further elaborate on the previous research on the topic of bearing performance of BIM-based safety risk assessment of nearshore tunnel to clarify it for readers.
While BIM is based on visualization of construction process, the manuscript seems to lack sufficient visualization and there is no graphical representation of the developed research.
It is necessary for the authors to provide robust verification/validation for this research. Given that the researchers used Revit to perform the analyses, a second conduction of analyses and performing pairwise comparisons and the following verification/validation of results is of utmost importance.
The authors may need to provide further justifications for their claims on the manuscript’s contribution to the body of knowledge. It is necessary for the authors to prove the innovation of this research and its main contribution to the body of knowledge.
The explanations around the specific evaluation criteria for this project's index levels and its relation to risk assessment need further elaboration.
The authors may need to further mention the limitations of their study in the conclusion section.
Moderate editing of English language required.
Reviewer 2 Report
The paper investigates an interesting topic, such as construction safety Risk Assessment and Early Warning of Nearshore Tunnel Based on BIM Technology. The methodology is appropriate and English is fine. However, some issues need to be considered.
Introduction
The novelties need to be expressed in detail in order to support the originality of the paper against the existiting literature.
The role of resilience on safety risk assessment needs to be discussed.
Section 2
This sentence: "In the contemporary world, continuous innovation in information technology is facilitating the deep development of digitalization, networking, and intelligence", needs to be expanded and explained.
The authors need to specify which are the capabilities of the platform in automatically detecting safety risk factors associated with scaffolding operations and devise appropriate risk mitigation measures.
Section 3
Is this part original? If not, it should be reduced with the due citations to the existing reference.
Tables 5-8 need to be explained in details
Conclusion
This part is more pertinent to a discussion. I suggest to change the title into: "Discussion" and a new conclusion (limitations, applications, future work...) needs to be written.
English is fine
Round 2
Reviewer 1 Report
Thank you for addressing the comments. The manuscript has been significantly improved after your revisions.
Minor editing of English language may required.
Author Response
We would like to express our gratitude to you for your thorough review of our article. Your feedback has been greatly appreciated, and we look forward to any additional input if it arises in the future. Thank you for your time and consideration.
Reviewer 2 Report
The role of resilience still needs to be discussed by referring to the due literature. Please refer to:
Cimellaro GP, Reinhorn AM, Bruneau M. Framework for analytical quantification of disaster resilience. Engineering Structures 2010;32:3639–49.
Forcellini D. An expeditious framework for assessing the seismic resilience (SR) of structural configurations; 56(2023): 105015.
Sangaki AH, Rofooei FR, Vafai H. Probabilistic integrated framework and models compatible with the reliability methods for seismic resilience assessment of structures. Structures 2021;34(2021):4086–99.
Round 3
Reviewer 2 Report
The paper is now ready for acceptance.